# Sequential Information Design:
# Learning to Persuade in the Dark

**Martino Bernasconi**[*]
Politecnico di Milano

**Matteo Castiglioni**[*]
Politecnico di Milano

**Alberto Marchesi**[*]
Politecnico di Milano

**Nicola Gatti**[*]
Politecnico di Milano

**Francesco Trovò**[*]
Politecnico di Milano

## Abstract

We study a repeated *information design* problem faced by an informed *sender* who tries to influence the behavior of a self-interested *receiver*. We consider settings where the receiver faces a *sequential decision making* (SDM) problem. At each round, the sender observes the realizations of random events in the SDM problem. This begets the challenge of how to incrementally disclose such information to the receiver to *persuade* them to follow (desirable) action recommendations. We study the case in which the sender does *not* know random events probabilities, and, thus, they have to gradually learn them while persuading the receiver. We start by providing a non-trivial polytopal approximation of the set of sender's persuasive information structures. This is crucial to design efficient learning algorithms. Next, we prove a negative result: no learning algorithm can be persuasive. Thus, we relax persuasiveness requirements by focusing on algorithms that guarantee that the receiver's regret in following recommendations grows sub-linearly. In the *full-feedback* setting—where the sender observes *all* random events realizations—, we provide an algorithm with $\tilde{O}(\sqrt{T})$ regret for both the sender and the receiver. Instead, in the *bandit-feedback* setting—where the sender only observes the realizations of random events actually occurring in the SDM problem—, we design an algorithm that, given an $\alpha \in [1/2, 1]$ as input, ensures $\tilde{O}(T^\alpha)$ and $\tilde{O}(T^{\max\{\alpha, 1-\frac{\alpha}{2}\}})$ regrets, for the sender and the receiver respectively. This result is complemented by a lower bound showing that such a regrets trade-off is essentially tight.

## 1 Introduction

Bayesian persuasion [Kamenica and Gentzkow, 2011] (a.k.a. *information design*) is the problem faced by an informed *sender* who wants to influence the behavior of a self-interested *receiver* via the provision of payoff-relevant information. This captures the problem of "who gets to know what", which is fundamental in all economic interactions. Thus, Bayesian persuasion is ubiquitous in real-world problems, such as, *e.g.*, online advertising [Bro Miltersen and Sheffet, 2012], voting [Alonso and Câmara, 2016, Castiglioni et al., 2020a, Castiglioni and Gatti, 2021], traffic routing [Bhaskar et al., 2016, Castiglioni et al., 2021a], security [Rabinovich et al., 2015, Xu et al., 2016], and marketing [Babichenko and Barman, 2017, Candogan, 2019].

We study Bayesian persuasion in settings where the receiver plays in a *sequential decision making* (SDM) problem. An SDM problem is characterized by a tree structure made by: *decision* nodes, where the receiver takes actions, and *chance* nodes, in which *partially observable* random events occur. The sender perfectly observes the realizations of random events, and their goal is to incrementally disclose

---

[*]Email:{martino.bernasconideluca,matteo.castiglioni,alberto.marchesi,nicola.gatti,francesco1.trovo}@polimi.it

the acquired information to induce the receiver towards desirable outcomes for the sender. This is not an easy feat when the sender and the receiver have different utilities. In order to do so, the sender commits to a *signaling scheme* specifying a probability distribution over action recommendations for the receiver at each decision node. Specifically, the sender commits to a *persuasive* signaling scheme, meaning that the receiver is incentivized to follow recommendations. We consider the case of a *farsighted* receiver, meaning that they take into account all the possible future events when deciding whether to deviate or *not* from recommendations at each decision node.

With some notable exceptions (such as, *e.g.*, [Zu et al., 2021]), Bayesian persuasion models in the literature make the stringent assumption that both the sender and the receiver know the *prior*, which, in our setting, is defined by the probabilities associated with random events in the SDM problem. We relax such an assumption by considering an online learning framework in which the sender, without any knowledge of the prior, repeatedly interacts with the receiver to gradually learn the prior while still being persuasive.

A concrete example of a setting that fits our model is that of a navigation app sending route recommendations to a driver. The app (sender) gets to know information about traffic congestion sequentially, since it might change over time, especially for long routes. Moreover, at any point in time, the app can send recommendations to the driver (receiver), who in turn has to take decisions on which routes to choose sequentially. The app and the driver might have different utilities. For example, the app may want to maximize overall traffic congestion, while the driver has to minimize their travel time. Moreover, the receiver interacts with the app multiple times and, thus, the application must provide good recommendations to the receiver, otherwise they would switch to another navigation app.

**Original contributions.** Our goal is to design online learning algorithms that are no-regret for the sender, while being persuasive for the receiver. We start by providing a non-trivial polytopal approximation of the set of sender's persuasive signaling schemes. This will be crucial in designing efficient (*i.e.*, polynomial-time) learning algorithms, and it also shows how a sender-optimal signaling scheme can be found in polynomial time in the offline version of our problem, which may be of independent interest. Next, we prove a negative result: without knowing the prior, no algorithm can be persuasive at each round with high probability. Thus, we relax persuasiveness requirements by focusing on learning algorithms that guarantee that the receiver's regret in following recommendations grows sub-linearly, while guaranteeing the same for sender's regret. First, we study the *full-feedback* case, where the sender observes the realizations of *all* the random events that may potentially happen in the SDM problem. In such a setting, we provide an algorithm with $\tilde{O}(\sqrt{T})$ regret for both the sender and the receiver. Then, we focus on the *bandit-feedback* setting, where the sender only observes the realizations of random events on the path in the tree traversed during the SDM problem. In this case, we design an algorithm that achieves $\tilde{O}(T^{\alpha})$ sender's regret and $\tilde{O}(T^{\max\{\alpha, 1-\frac{\alpha}{2}\}})$ receiver's regret, for any $\alpha \in [1/2, 1]$ given as input. The crucial component of the algorithm is a non-trivial exploration phase that uniformly explores the tree defining the SDM problem to build suitable estimators of the prior. This is needed since, with bandit feedback, playing a signaling scheme may provide insufficient information about its persuasiveness. Finally, we provide a lower bound showing that the regrets trade off achieved by our algorithm is tight for $\alpha \in [1/2, 2/3]$.

**Related works.** Some works addressed Bayesian persuasion in *Markov decision processes* (MDPs). Gan et al. [2022] and Wu et al. [2022] show how to efficiently find a sender-optimal policy when the receiver is *myopic* (*i.e.*, it only optimizes one-step rewards) in MDPs with infinite and finite horizon, respectively. Moreover, the former assume that the environment is known, while the latter do *not*. These works considerably differ from ours, since we assume a farsighted receiver and also model partial observability of random events.[2] Another work close to ours is [Zu et al., 2021], which studies a (non-sequential) persuasion problem in which the sender and the receiver do *not* know the prior and interact online. Zu et al. [2021] provide a persuasive learning algorithm, while, in our model, we show that the ignorance of the prior precludes the possibility of committing to persuasive signaling schemes, and, thus, we need to resort to new techniques to circumvent the issue. Another line of research, that uses similar techniques as the one employed in this work, studies learning in SDM problems while satisfying unknown constraints [Bernasconi-de-Luca et al., 2021, Bernasconi et al., 2022]. Finally, Celli et al. [2020a] study Bayesian persuasion with multiple receivers interacting in an

---

[2]Gan et al. [2022] also study a model with farsighted receiver, where they show that the problem of finding a sender-optimal policy is NP-hard. Thus, they do *not* provide any algorithmic result for such a model.

imperfect-information sequential game. Differently from ours, their model adopts a different notion of persuasiveness, known as *ex ante* persuasiveness, and it assumes that the prior is known. Other works study learning problems in which the sender does *not* know the receivers' payoffs (but knows the prior); see, *e.g.*, [Castiglioni et al., 2020b, 2021b, 2022].

## 2 Preliminaries

### 2.1 Sequential decision making problems

An instance of an SDM problem is defined by a tree structure, utilities, and random events probabilities. The tree structure has a set of nodes $\mathcal{H} := \mathcal{Z} \cup \mathcal{H}_d \cup \mathcal{H}_c$, where: $\mathcal{Z}$ contains all the *terminal nodes* in which the problem ends (corresponding to the leaves of the tree), $\mathcal{H}_d$ is the set of *decision nodes* in which the agent acts, while $\mathcal{H}_c$ is the set of *chance nodes* where random events occur. Given any non-terminal node $h \in \mathcal{H} \setminus \mathcal{Z}$, we let $A(h)$ be the set of arcs outgoing from $h$. If $h \in \mathcal{H}_d$, then $A(h)$ is the set of receiver's actions available at $h$, while, if $h \in \mathcal{H}_c$, then $A(h)$ encodes the possible outcomes of the random event occurring at $h$. Furthermore, the utility function $u : \mathcal{Z} \to [0, 1]$ defines the agent's payoff $u(z)$ when the problem ends in terminal node $z \in \mathcal{Z}$. Finally, each chance node $h \in \mathcal{H}_c$ is characterized by a probability distribution $\mu_h \in \Delta_{A(h)}$ over the possible outcomes of the corresponding random event, with $\mu_h(a)$ denoting the probability of action $a \in A(h)$.[3]

In an SDM problem, the agent has *imperfect information*, since they do *not* perfectly observe the outcomes of random events. Thus, the set of decision nodes $\mathcal{H}_d$ is partitioned into *information sets* (infosets for short), where an infoset $I \subseteq \mathcal{H}_d$ is a subset of decision nodes that are indistinguishable for the agent. We denote the set of infosets as $\mathcal{I}$. For every infoset $I \in \mathcal{I}$ and pair of nodes $h, h' \in I$, it must be the case that $A(h) = A(h') =: A(I)$, otherwise the agent could distinguish between the two nodes. We assume that the agent has *perfect recall*, which means that they never forget information once acquired. Formally, this is equivalent to assume that, for every infoset $I \in \mathcal{I}$, all the paths from the root of the tree to a node $h \in I$ identify the same ordered sequence of agent's actions.

### 2.2 Bayesian persuasion in sequential decision making problems

We study *Bayesian persuasion in SDM* (BPSDM) problems. These extend the Bayesian persuasion framework [Kamenica and Gentzkow, 2011] to SDM problems by introducing an exogenous agent that acts as a *sender* by issuing signals to the decision-making agent (the *receiver*).[4] By following the Bayesian persuasion terminology, the probability distributions $\mu_h$ for each chance node $h$ are collectively referred to as the *prior*. Thus, the sender observes the realizations of random events occurring in the SDM problem and can partially disclose information to influence the receiver's behavior. Moreover, the sender has their own utility function defined over terminal nodes, denoted as $f : \mathcal{Z} \to [0, 1]$, and their goal is to commit to a publicly known *signaling scheme* that maximizes their utility in expectation with respect to the prior, the signaling scheme, and the receiver's strategy.

Formally, a signaling scheme for the sender defines a probability distribution $\phi_h \in \Delta_{S(h)}$ at each decision node $h \in \mathcal{H}_d$, where $S(h)$ is a finite set of signals available at $h$. During the SDM problem, when the receiver reaches a node $h \in \mathcal{H}_d$ belonging to an infoset $I \in \mathcal{I}$, the sender draws a signal $s \sim \phi_h$ and communicates it to the receiver. Then, based on the history of signals observed from the beginning of the SDM problem ($s$ included), the receiver computes a *posterior* belief over the nodes belonging to the infoset $I$ and plays so as to maximize their expected utility in the SDM sub-problem that starts from $I$, taking into account the just acquired information.

As customary in these settings, a simple revelation-principle-style argument allows us to focus on signaling schemes that are *direct* and *persuasive* [Kamenica and Gentzkow, 2011, Arieli and Babichenko, 2019]. In particular, a signaling scheme is direct if signals correspond to action recommendations, namely $S(h) = A(h)$ for all $h \in \mathcal{H}_d$. A direct signaling scheme is persuasive if the receiver is incentivized to follow action recommendations issued by the sender. Moreover, we assume that, if the receiver does *not* follow action recommendations at some decision node, then the sender stops issuing recommendations at nodes later reached during the SDM problem. This

---

[3]For a finite set $X$ we denote with $\Delta_X$ the set of probability distributions over $X$.

[4]Appendix A shows that BPSDM reduces to classical Bayesian persuasion when there is no sequentiality.

is without loss of generality. We refer to [Von Stengel and Forges, 2008, Morrill et al., 2021] for a discussion on a similar problem in the field of correlation in sequential games.

## 2.3 The sequence-form representation

The *sequence form* is a commonly-used, compact way of representing *(mixed) strategies* in SDM problems [Koller et al., 1996]. In this work, the sequence-form representation will be employed for receiver's strategies, and to encode the signaling schemes and priors, as we describe in the following.

**Receiver's strategies.** Given any $h \in \mathcal{H}$, we let $\sigma_r(h)$ be the ordered *sequence* of receiver's actions on the path from the root of the tree to node $h$. By the perfect recall assumption, given any infoset $I \in \mathcal{I}$, it holds that $\sigma_r(h) = \sigma_r(h') =: \sigma_r(I)$ for every pair of nodes $h, h' \in I$. Thus, we can identify sequences with infoset-action pairs, with $\sigma = (I, a)$ encoding the sequence of actions obtained by appending action $a \in A(I)$ at the end of $\sigma_r(I)$, for any infoset $I \in \mathcal{I}$. Moreover, $\varnothing$ denotes the *empty sequence*. Hence, the receiver's sequences are $\Sigma_r := \{(I, a) \mid I \in \mathcal{I}, a \in A(I)\} \cup \{\varnothing\}$. In the sequence-form representation, mixed strategies are defined by specifying the probability of playing each sequence of actions. Thus, a receiver's strategy is represented by a vector $\boldsymbol{x} \in [0, 1]^{|\Sigma_r|}$, where $\boldsymbol{x}[\sigma]$ encodes the realization probability of sequence $\sigma \in \Sigma_r$. Furthermore, a sequence-form strategy is well-defined if and only if it satisfies the following linear constraints:

$$\boldsymbol{x}[\varnothing] = 1 \quad \text{and} \quad \boldsymbol{x}[\sigma_r(I)] = \sum_{a \in A(I)} \boldsymbol{x}[\sigma_r(I)a] \quad \forall I \in \mathcal{I}.$$

We denote by $\mathcal{X}_r$ the polytope of all receiver's sequence-form strategies. We will also need to work with the sets of receiver's strategies in the SDM sub-problem that starts from an infoset $I \in \mathcal{I}$, formally defined as $\mathcal{X}_{r,I} := \{\boldsymbol{x} \in \mathcal{X}_r \mid \boldsymbol{x}[\sigma_r(I)] = 1\}$.

**Signaling schemes.** We represent signaling schemes in sequence form by leveraging the fact that the sender can be thought of as a perfect-information agent who plays at the decision nodes of the SDM problem, since their actions correspond to recommendations for the receiver. Thus, since sender's infosets correspond to decision nodes, their sequences $\Sigma_s := \{(h, a) \mid h \in \mathcal{H}_d, a \in A(h)\} \cup \{\varnothing\}$. Then, we denote the polytope of (sequence-form) signaling schemes as $\Phi \subseteq [0, 1]^{|\Sigma_s|}$, where each signaling scheme is represented as a vector $\boldsymbol{\phi} \in [0, 1]^{|\Sigma_s|}$ satisfying:

$$\boldsymbol{\phi}[\varnothing] = 1 \quad \text{and} \quad \boldsymbol{\phi}[\sigma_s(h)] = \sum_{a \in A(h)} \boldsymbol{\phi}[\sigma_s(h)a] \quad \forall h \in \mathcal{H}_d,$$

where, similarly to $\sigma_r(h)$ for the receiver, $\sigma_s(h)$ denotes the sender's sequence identified by $h \in \mathcal{H}$. We also define $\Pi := \Phi \cap \{0, 1\}^{|\Sigma_s|}$ as the set of *deterministic* signaling schemes, which are those that recommend a single action with probability one at each decision node.

**Priors.** We also encode prior probability distributions $\mu_h$ by means of the sequence form. Indeed, these can be though of as elements of a fixed strategy played by a (fictitious) perfect-information agent that acts at chance nodes. Thus, for such a chance agent, we define $\Sigma_c$, $\mathcal{X}_c$, and $\sigma_c(h)$ as their counterparts previously introduced for the receiver. Moreover, in the following, we denote by $\boldsymbol{\mu}^\star \in \mathcal{X}_c$ the (sequence-form) prior, recursively defined as follows:

$$\boldsymbol{\mu}^\star[\varnothing] := 1 \quad \text{and} \quad \boldsymbol{\mu}^\star[\sigma_c(h)a] := \boldsymbol{\mu}^\star[\sigma_c(h)] \, \mu_h(a) \quad \forall h \in \mathcal{H}_c, \forall a \in A(h).$$

**Ordering of sequences.** For the sake of presentation, we introduce a partial ordering relation among sequences. Given two sequences $\sigma = (I, a) \in \Sigma_r$ and $\sigma' = (J, b) \in \Sigma_r$, we write $\sigma \preceq \sigma'$ (read as $\sigma$ *precedes* $\sigma'$), whenever there exists a path in the tree connecting a node in $I$ to a node in $J$, and such a path includes action $a$. We adopt analogous definitions for sequences in $\Sigma_s$ and $\Sigma_c$.[5]

## 3 Learning to persuade

In this work, we relax the strong assumption that both the sender and the receiver know the prior $\boldsymbol{\mu}^\star$ by casting the BPSDM problem into an *online learning framework* in which the sender repeatedly interacts with the receiver over a time horizon of length $T$. At each round $t \in [T]$, the interaction

---

[5]We refer the reader to Appendix B for an example of SDM problem and its sets of sequences.

goes as follows:[6] (i) the sender commits to a signaling scheme $\phi_t \in \Phi$; (ii) a vector $\boldsymbol{y}_t \in \{0, 1\}^{|\Sigma_c|}$ encoding realizations of random events is drawn according to $\boldsymbol{\mu}^\star$; (iii) the sender and the receiver play an instance of the (one-shot) BPSDM problem (detailed in Section 2.2), in which the sender commits to $\phi_t$, random events at chance nodes are realized as defined by $\boldsymbol{y}_t$, and the receiver sticks to the recommendations issued by the sender; and (iv) the sender observes a *feedback* on the realization of random events at chance nodes, which can be of two types: *full feedback* when the sender observes $\boldsymbol{y}_t$, which specifies the realizations of *all* the random events at chance nodes that are possibly reachable during the SDM problem; *bandit feedback* when the sender observes the terminal node $z_t \in \mathcal{Z}$ reached at the end of the SDM problem. The latter is equivalent to observing the realizations of random events at the chance nodes that are actually reached during the SDM problem, namely $\sigma_c(z_t)$.

By letting $\Phi^\diamond(\boldsymbol{\mu}^\star)$ be the set of persuasive signaling schemes, *i.e.*, such that the receiver is incentivized to following recommendations (a formal definition is provided in Definition 2), the goal of the sender is to select a sequence of signaling schemes, namely $\phi_1, \ldots, \phi_T$, which maximizes their expected utility, while guaranteeing that each signaling scheme $\phi_t$ is persuasive, namely $\phi_t \in \Phi^\diamond(\boldsymbol{\mu}^\star)$.

We measure the performance of a sequence $\phi_1, \ldots, \phi_T$ of signaling schemes by comparing it with an optimal (fixed) persuasive signaling scheme. Formally, given a signaling scheme $\phi \in \Phi$, we first define $U(\phi, \boldsymbol{\mu}^\star)$, respectively $F(\phi, \boldsymbol{\mu}^\star)$, as the expected utility achieved by the receiver, respectively the sender, whenever the former follows action recommendations. These can be expressed as linear functions of $\phi$, which, for any $\boldsymbol{\mu} \in \mathcal{X}_c$, are defined as follows:

$$U(\phi, \boldsymbol{\mu}) := \sum_{z \in \mathcal{Z}} \boldsymbol{\mu}[\sigma_c(z)]\phi[\sigma_s(z)]u(z), \quad F(\phi, \boldsymbol{\mu}) := \sum_{z \in \mathcal{Z}} \boldsymbol{\mu}[\sigma_c(z)]\phi[\sigma_s(z)]f(z).$$

Finally, by letting $\phi^\star \in \operatorname{argmax}_{\phi \in \Phi^\diamond(\boldsymbol{\mu}^\star)} F(\phi, \boldsymbol{\mu}^\star)$ be an optimal (fixed) persuasive signaling scheme, the sender' performance over $T$ rounds is measured by the *(cumulative) sender's regret*:

$$R_T := \sum_{t \in [T]} \left( F(\phi^\star, \boldsymbol{\mu}^\star) - F(\phi_t, \boldsymbol{\mu}^\star) \right).$$

The goal is to design learning algorithms (for the sender) which select sequences of persuasive signaling schemes such that $R_T$ grows asymptotically sub-linearly in $T$, namely $R_T = o(T)$.

## 4 On the characterization of persuasive signaling schemes

### 4.1 A local decomposition of persuasiveness

In this section, we formally introduce the set of persuasive signaling schemes $\Phi^\diamond(\boldsymbol{\mu}^\star)$ as the set of signaling schemes for which the receiver's expected utility by following recommendations is greater than the one provided by an optimal *deviation policy* (DP).[7] In addition, we show how to decompose any DP into components defined locally at each infoset, which will be crucial in the following Section 4.2. Intuitively, a DP for the receiver is specified by two elements: (i) a set of *deviation points* in which the DP prescribes to stop following action recommendations; and (ii) the *continuation strategies* to be adopted after deviating from recommendations.

We represent deviation points by vectors $\boldsymbol{\omega} \in \{0, 1\}^{|\Sigma_r|}$, which are defined so that $\boldsymbol{\omega}[\sigma] = 1$ if and only if the DP prescribes to deviate upon observing the sequence of action recommendations $\sigma \in \Sigma_r$. Moreover, by leveraging the w.l.o.g. assumption that the sender stops issuing recommendations after the receiver deviated from them, we focus on DPs such that each path from the root of the tree to a terminal node involves only one deviation point. As a result, the set of all valid vectors $\boldsymbol{\omega} \in \{0, 1\}^{|\Sigma_r|}$ is formally defined as $\Omega := \left\{ \boldsymbol{\omega} \in \{0, 1\}^{|\Sigma_r|} \mid \sum_{\sigma \in \Sigma_r : \sigma \preceq \sigma_r(z)} \boldsymbol{\omega}[\sigma] \leq 1 \quad \forall z \in \mathcal{Z} \right\}$.

We represent the continuation strategies of DPs by introducing the set of *continuation strategy profiles*, denoted as $\mathcal{P} := \bigtimes_{\sigma = (I,a) \in \Sigma_r} \mathcal{X}_{r,I}$. A continuation strategy profile $\boldsymbol{\rho} \in \mathcal{P}$, with $\boldsymbol{\rho} = (\boldsymbol{\rho}_\sigma)_{\sigma \in \Sigma_r}$, defines a strategy $\boldsymbol{\rho}_\sigma \in \mathcal{X}_{r,I}$ for every receiver's sequence $\sigma = (I, a) \in \Sigma_r$. Intuitively, $\boldsymbol{\rho}_\sigma$ is the strategy for the SDM sub-problem starting from infoset $I$ that is used by the receiver after deviating upon observing sequence $\sigma \in \Sigma_r$. As a result, any pair $(\boldsymbol{\omega}, \boldsymbol{\rho}) \in \Omega \times \mathcal{P}$ specifies a valid DP; formally:

---

[6] Throughout this work, for $n \in \mathbb{N}$, we denote with $[n]$ the set $\{1, \ldots, n\}$.

[7] For ease of exposition, all the definitions and results in this section are provided for the prior $\boldsymbol{\mu}^\star$. It is straightforward to generalize them to the case of a generic $\boldsymbol{\mu} \in \mathcal{X}_c$.

**Definition 1** (Deviation policy). *Given a vector $\boldsymbol{\omega} \in \Omega$ and a profile $\boldsymbol{\rho} \in \mathcal{P}$, the $(\boldsymbol{\omega}, \boldsymbol{\rho})$-DP prescribes to follow sender's recommendations until action $a$ is recommended at infoset $I$ for some sequence $\sigma = (I, a)$ such that $\boldsymbol{\omega}[\sigma] = 1$; from that point on, it prescribes to play according to strategy $\boldsymbol{\rho}_\sigma$.*

We denote by $U^{\boldsymbol{\omega} \to \boldsymbol{\rho}}(\boldsymbol{\phi}, \boldsymbol{\mu}^\star)$ the receiver's expected utility obtained with a $(\boldsymbol{\omega}, \boldsymbol{\rho})$-DP, so that we can state the following formal definition of persuasive signaling schemes.

**Definition 2** (Persuasiveness). *A signaling scheme $\boldsymbol{\phi} \in \Phi$ is $\epsilon$-persuasive, namely $\boldsymbol{\phi} \in \Phi_\epsilon^\diamond(\boldsymbol{\mu}^\star)$, if*

$$\max_{(\boldsymbol{\omega}, \boldsymbol{\rho}) \in \Omega \times \mathcal{P}} U^{\boldsymbol{\omega} \to \boldsymbol{\rho}}(\boldsymbol{\phi}, \boldsymbol{\mu}^\star) - U(\boldsymbol{\phi}, \boldsymbol{\mu}^\star) \leq \epsilon. \tag{1}$$

*Moreover, a signaling scheme $\boldsymbol{\phi} \in \Phi$ is* persuasive, *namely $\boldsymbol{\phi} \in \Phi^\diamond(\boldsymbol{\mu}^\star)$, if it is 0-persuasive.*

Intuitively, the above definition states that a signaling scheme is $\epsilon$-persuasive if the receiver's expected utility by following recommendations is at most $\epsilon$ less than the one obtained by an optimal DP, which is a DP maximizing receiver's expected utility.

Our local decomposition of DPs is based on suitably-defined, simple deviation policies, which we call *single-point DPs* (SPDPs). These are a special case of DPs that stop following sender's action recommendations only when a specific single infoset is reached and a particular action is recommended therein. SPDPs are formally defined as follows:

**Definition 3** (Single-point deviation strategy). *Given a receiver's sequence $\sigma = (I, a) \in \Sigma_r$ and a receiver's strategy $\boldsymbol{\rho}_\sigma \in \mathcal{X}_{r,I}$ for the SDM sub-problem starting from infoset $I$, the $(\sigma, \boldsymbol{\rho}_\sigma)$-SPDP prescribes to follow sender's recommendations until action $a$ is recommended at infoset $I$; from that point on, the strategy prescribes to play according to $\boldsymbol{\rho}_\sigma$.*

We denote by $U_{\sigma \to \boldsymbol{\rho}_\sigma}(\boldsymbol{\phi}, \boldsymbol{\mu}^\star)$ the receiver's expected utility obtained by following an $(\sigma, \boldsymbol{\rho}_\sigma)$-SPDP. The following theorem provides the key result underlying our decomposition.[8] It shows that the difference between the utility achieved by a $(\boldsymbol{\omega}, \boldsymbol{\rho})$-DP and that obtained by following recommendations can be decomposed into the sum over all the sequences $\sigma \in \Sigma_r$ of analogous differences defined for the $(\sigma, \boldsymbol{\rho}_\sigma)$-SPDPs, where each difference is weighted by $\boldsymbol{\omega}[\sigma]$.

**Theorem 1.** *Given a signaling scheme $\boldsymbol{\phi} \in \Phi$ and a $(\boldsymbol{\omega}, \boldsymbol{\rho})$-DP, it holds:*

$$U^{\boldsymbol{\omega} \to \boldsymbol{\rho}}(\boldsymbol{\phi}, \boldsymbol{\mu}^\star) - U(\boldsymbol{\phi}, \boldsymbol{\mu}^\star) = \sum_{\sigma \in \Sigma_r} \boldsymbol{\omega}[\sigma] \Big( U_{\sigma \to \boldsymbol{\rho}_\sigma}(\boldsymbol{\phi}, \boldsymbol{\mu}^\star) - U(\boldsymbol{\phi}, \boldsymbol{\mu}^\star) \Big).$$

### 4.2 A polytopal approximation of the set of persuasive signaling schemes

In the following, we show how to exploit Theorem 1 to provide an approximate characterization of the set $\Phi_\epsilon^\diamond(\boldsymbol{\mu}^\star)$ using a polynomially-sized polytope. First, we state a corollary of Theorem 1 showing that persuasiveness can be bounded by suitably defined SPDPs. Formally:[9]

**Corollary 1.** *Given a signaling scheme $\boldsymbol{\phi} \in \Phi$, the following holds:*

$$\max_{(\boldsymbol{\omega}, \boldsymbol{\rho}) \in \Omega \times \mathcal{P}} U^{\boldsymbol{\omega} \to \boldsymbol{\rho}}(\boldsymbol{\phi}, \boldsymbol{\mu}^\star) - U(\boldsymbol{\phi}, \boldsymbol{\mu}^\star) \leq \sum_{\sigma = (I, a) \in \Sigma_r} \left[ \max_{\boldsymbol{\rho}_\sigma \in \mathcal{X}_{r,I}} U_{\sigma \to \boldsymbol{\rho}_\sigma}(\boldsymbol{\phi}, \boldsymbol{\mu}^\star) - U(\boldsymbol{\phi}, \boldsymbol{\mu}^\star) \right]^+.$$

By exploiting Corollary 1, we introduce the following definition of $\epsilon$-*persuasive polytope* (Lemma 1 justifies the term polytope), as the set of signaling schemes for which there is no $(\sigma, \boldsymbol{\rho}_\sigma)$-SPDP that achieves a receiver's utility that exceeds by more than $\epsilon / |\Sigma_r|$ that of following recommendations.

**Definition 4** (Persuasive polytope). *The $\epsilon$-persuasive polytope is defined as:*

$$\Lambda_\epsilon(\boldsymbol{\mu}^\star) := \left\{ \boldsymbol{\phi} \in \Phi \; \Big| \; \max_{\boldsymbol{\rho}_\sigma \in \mathcal{X}_{r,I}} U_{\sigma \to \boldsymbol{\rho}_\sigma}(\boldsymbol{\phi}, \boldsymbol{\mu}^\star) - U(\boldsymbol{\phi}, \boldsymbol{\mu}^\star) \leq \epsilon / |\Sigma_r| \quad \forall \sigma \in \Sigma_r \right\}.$$

*Moreover, we denote by $\Lambda(\boldsymbol{\mu}^\star)$ the 0-persuasive polytope.*

As we show in the following lemma, $\Lambda_\epsilon(\boldsymbol{\mu}^\star)$ is an efficiently-representable polytope.

**Lemma 1.** *The set $\Lambda_\epsilon(\boldsymbol{\mu}^\star)$ can be described by means of a polynomial number of linear constraints.*

---

[8] All the proofs are provided in the Appendices D, E, F, and G.

[9] Given any $x \in \mathbb{R}$, we let $[x]^+ := \max(x, 0)$.

The following lemma shows that the $\epsilon$-persuasive polytope is contained in $\Phi_\epsilon^\diamond(\boldsymbol{\mu}^\star)$, and that the set of persuasive signaling schemes is contained in the former. Formally:

**Lemma 2.** *It is always the case that $\Phi^\diamond(\boldsymbol{\mu}^\star) \equiv \Lambda(\boldsymbol{\mu}^\star) \subseteq \Lambda_\epsilon(\boldsymbol{\mu}^\star) \subseteq \Phi_\epsilon^\diamond(\boldsymbol{\mu}^\star)$.*

Lemma 2 also implies that the polytope $\Lambda(\boldsymbol{\mu}^\star)$ exactly characterizes the set of persuasive signaling schemes $\Phi^\diamond(\boldsymbol{\mu}^\star)$. Thus, by adding the maximization of the sender's expected utility $F(\boldsymbol{\phi}, \boldsymbol{\mu}^\star)$ on top of the linear constraints describing $\Lambda(\boldsymbol{\mu}^\star)$, we obtain a polynomially-sized linear program for finding an optimal sender's signaling scheme in any instance of the BPSDM problem in which $\boldsymbol{\mu}^\star$ is known.

**Theorem 2.** *The BPSDM problem can be solved in polynomial time when the prior $\boldsymbol{\mu}^\star$ is known.*

## 5   Always being persuasive is impossible: a relaxation is needed

In this section, we prove that it is impossible to design an algorithm that returns a sequence of persuasive signaling schemes for a generic BPSDM problem.

**Theorem 3** (Impossibility of persuasiveness). *There exists a constant $\gamma \in (0,1)$ such that no algorithm can guarantee to output a sequence $\boldsymbol{\phi}_1, \ldots, \boldsymbol{\phi}_T$ of signaling schemes such that, with probability al least $\gamma$, all the signaling schemes $\boldsymbol{\phi}_t$ are persuasive.*

Notice that this result is in contrast with what happens in non-sequential Bayesian persuasion (see the work by Zu et al. [2021]), where it is possible to design no-regret algorithms that output sequences of signaling schemes that are guaranteed to be persuasive with high probability. Theorem 3 motivates the introduction of a less restrictive requirement on the signaling schemes. In particular, we look for algorithms that output signaling schemes $\boldsymbol{\phi}_1, \ldots, \boldsymbol{\phi}_T$, such that the expected utility loss incurred by the receiver by following recommendations rather than playing an optimal DP is small. To capture such a requirement, we introduce the following definition of *(cumulative) receiver's regret*:

$$V_T := \sum_{t\in[T]} \max_{(\boldsymbol{\omega},\boldsymbol{\rho})\in\Omega\times\mathcal{P}} U^{\boldsymbol{\omega}\to\boldsymbol{\rho}}(\boldsymbol{\phi}_t, \boldsymbol{\mu}^\star) - \sum_{t\in[T]} U(\boldsymbol{\phi}_t, \boldsymbol{\mu}^\star).$$

Therefore our goal becomes that of designing algorithms guaranteeing that the cumulative receiver's regret grows sub-linearly in $T$, namely $V_T = o(T)$, while continuing to ensure that $R_T = o(T)$.

In Sections 6 and 7, we design algorithms achieving sub-linear $V_T$ and $R_T$ for the learning problem described in Section 3. The algorithms implement two functions: (i) SELECTSTRATEGY(), which, at each $t \in [T]$, draws a signaling scheme $\boldsymbol{\phi}_t \in \Phi$ on the basis of the internal state of the algorithm; and (ii) UPDATE($o_t$), which modifies the internal state on the basis of the observation $o_t$ received as feedback. Each algorithm alternates these two functions as the interaction between the sender and the receiver unfolds as described in Section 3. Specifically, under full feedback the sender observes $\boldsymbol{y}_t$ and calls UPDATE($\boldsymbol{y}_t$), while in the bandit feedback it observes $z_t$ and calls UPDATE($z_t$).

## 6   Learning with full feedback

In this section, we will discuss the online problem faced by the sender that wants to optimize online its utility $F(\boldsymbol{\phi}, \boldsymbol{\mu}^\star)$ while learning the unknown prior $\boldsymbol{\mu}^\star$. We start by providing a learning algorithm (Algorithm 1) working with full feedback, *i.e.*, when the sender observes the realizations of *all* the possible random events. The main idea of the algorithm is to choose signaling schemes $\boldsymbol{\phi}_t$ that belong to suitable sets $\Lambda_{\beta_t}(\widehat{\boldsymbol{\mu}}_t)$ which are designed to be "close" to the set $\Phi^\diamond(\boldsymbol{\mu}^\star)$ of persuasive signaling schemes. At each round $t \in [T]$, Algorithm 1 defines the desired set as follows. First, it maintains an estimate $\widehat{\boldsymbol{\mu}}_t$ of $\boldsymbol{\mu}^\star$; formally, it defines a radius $\epsilon_t$ such that the event $\mathcal{E} := \{\|\widehat{\boldsymbol{\mu}}_t - \boldsymbol{\mu}^\star\|_\infty \leq \epsilon_t \ \forall t \in [T]\}$ holds with probability at least $1 - \delta$. Second, it defines a parameter

---

**Algorithm 1** Full-feedback algorithm

**function** SELECTSTRATEGY():

$\quad \boldsymbol{\phi}_t \leftarrow \arg \max\limits_{\boldsymbol{\phi}\in\Lambda_{\beta_t}(\widehat{\boldsymbol{\mu}}_t)} F(\boldsymbol{\phi}, \widehat{\boldsymbol{\mu}}_t)$

$\quad$ **return** $\boldsymbol{\phi}_t$

---

**function** UPDATE($\boldsymbol{y}_t$):

$\quad \widehat{\boldsymbol{\mu}}_{t+1}[\sigma] \leftarrow \sum\limits_{\tau=1}^{t} \boldsymbol{y}_\tau[\sigma]/\tau \ \forall\sigma\in\Sigma_c$

$\quad \epsilon_{t+1} \leftarrow \sqrt{\frac{\log(2T|\Sigma_c|/\delta)}{2t}}$

$\quad \beta_{t+1} \leftarrow 2|\mathcal{Z}||\Sigma_r|\epsilon_{t+1}$

---

$\beta_t$ such that, conditionally to the realization of the event $\mathcal{E}$, the following two conditions hold: (i) the decision space $\Lambda_{\beta_t}(\widehat{\boldsymbol{\mu}}_t)$ contains the optimal signaling scheme $\boldsymbol{\phi}^\star$; (ii) $\Lambda_{2\beta_t}(\boldsymbol{\mu}^\star)$ contains the

signaling scheme $\phi_t$. Intuitively, the first condition is needed to have low sender's regret, while the second one yields signaling schemes that are approximately persuasive.[10]

The polytopal approximation that we provide in Section 4.2 plays a crucial role in the complexity of Algorithm 1. Specifically, it allows it to select the desired $\phi_t$ in polynomial time by optimizing over the set $\Lambda_{\beta_t}(\widehat{\boldsymbol{\mu}}_t)$, which can be done efficiently. The use of the set $\Lambda_{\beta_t}(\widehat{\boldsymbol{\mu}}_t)$ over $\Phi_{\beta_t}^{\diamond}(\widehat{\boldsymbol{\mu}}_t)$ is necessary due to the fact that the latter is *not* known to admit an efficient representation. Formally:

**Theorem 4.** *Given any $\delta \in (0,1)$, with probability at least $1 - \delta$, Algorithm 1 guarantees:*

$$R_T = \mathcal{O}\left(|\mathcal{Z}|\sqrt{T \log\left(T|\Sigma_c|/\delta\right)}\right), \quad V_T = \mathcal{O}\left(|\Sigma_r||\mathcal{Z}|\sqrt{T \log\left(T|\Sigma_c|/\delta\right)}\right).$$

*Moreover, the algorithm runs in polynomial time.*

## 7 Learning with bandit feedback

In this section, we build on Algorithm 1 to deal with bandit feedback, *i.e.*, when at each round $t \in [T]$ the sender only observes the terminal node $z_t$ reached at the end of the SDM problem. The main difficulties of such a setting can be summarized by the following observations. First, the feedback $z_t$ only reveals partial information about the prior, and such information also depends on the selected signaling scheme $\phi_t$. Second, even if the sender plays a signaling scheme $\phi \in \Phi$ for an arbitrarily large number of rounds, there is no guarantee that they collect enough information to tell whether $\phi \in \Phi_\epsilon^{\diamond}(\boldsymbol{\mu}^\star)$ or *not* for some $\epsilon > 0$. Indeed, the persuasiveness of a signaling scheme depends on *all* receiver's utilities in the SDM problem, and some parts of the tree may *not* be reached during a sufficiently large number of rounds by committing to $\phi$. Thus, any algorithm for the bandit-feedback setting must guarantee a suitable level of exploration over the entire tree, so as to keep track of the entity of the violation of persuasiveness constraints.

---

**Algorithm 2** Bandit-feedback algorithm

**function** SELECTSTRATEGY():
    **if** $t \leq N$ **then**          ▷ First Phase
        $\sigma = (h, a) \leftarrow \arg\min_{\sigma \in \Sigma_c} C_t[\sigma]$
        $\Sigma_s \ni \sigma' \leftarrow \sigma_s(h)$
        Choose $\phi_t \in \Phi : \phi_t[\sigma'] = 1$
    **else**          ▷ Second Phase
        $\phi_t \leftarrow \arg\max\limits_{\phi \in \Lambda_{\beta_N}(\widehat{\boldsymbol{\mu}}_N)} \max\limits_{\boldsymbol{\mu} \in \mathcal{C}_t(\delta)} F(\phi, \boldsymbol{\mu})$

    **return** $\phi_t$

---

**function** UPDATE($z_t$):
    Build path $\boldsymbol{p}_t \in \{0,1\}^{|\Sigma_c|}$ from $\sigma_c(z_t)$
    Sample $\boldsymbol{\pi}_t \sim \phi_t$ s.t. $\boldsymbol{p}_t[\sigma] = 1 \Rightarrow \sigma \in \Sigma_\downarrow(\boldsymbol{\pi}_t)$
    **for** $\sigma \in \Sigma_\downarrow(\boldsymbol{\pi}_t)$ **do**
        $C_{t+1}[\sigma] \leftarrow C_t[\sigma] + 1$
        $\widehat{\boldsymbol{\mu}}_{t+1}[\sigma] \leftarrow \frac{1}{C_{t+1}[\sigma]} \sum_{\tau=1}^{C_{t+1}[\sigma]} \boldsymbol{p}_\tau[\sigma]$
        $\epsilon_{t+1}[\sigma] \leftarrow \sqrt{\frac{\log(4T|\Sigma_c|/\delta)}{2C_{t+1}[\sigma]}}$
    $\mathcal{C}_{t+1}(\delta) \leftarrow \left\{ \boldsymbol{\mu} \,\middle|\, |\boldsymbol{\mu}[\sigma] - \widehat{\boldsymbol{\mu}}_{t+1}[\sigma]| \leq \epsilon_{t+1}[\sigma] \,\forall \sigma \in \Sigma_c \right\}$
    $\beta_{t+1} \leftarrow 2|\mathcal{Z}||\Sigma_c|\sqrt{\frac{|\Sigma_c|\log(4T|\Sigma_c|/\delta)}{2(t+1)}}$

---

We design a two-phase algorithm, whose pseudo-code is provided in Algorithm 2. The algorithm takes as input the number $N \in [T]$ of rounds devoted to the *first phase* guaranteeing the necessary amount of exploration, as detailed in Section 7.1. During this phase, the SELECTSTRATEGY() procedure implements an efficient deterministic uniform exploration policy, which builds an unbiased estimator $\widehat{\boldsymbol{\mu}}_N$ of $\boldsymbol{\mu}^\star$. This allows to restrict the space of feasible signaling schemes used in the subsequent phase to those that are approximately persuasive, *i.e.*, those in the set $\Lambda_{\beta_N}(\widehat{\boldsymbol{\mu}}_N)$. In Section 7.2, we discuss the *second phase* of the the algorithm, composed by the rounds $t > N$, during which the algorithm focuses on the minimization of sender's regret by exploiting the *optimism in face of uncertainty* principle. Finally, in Section 7.3, we provide a lower bound on the trade-off between sender's and receiver's regrets, matching the upper bounds achieved by Algorithm 2 for a large portion of the trade-off frontier. This result formally motivates the necessity of the uniform exploration which is performed in the first phase of the algorithm.

### 7.1 Minimizing the receiver's regret

At each round $t \in [T]$, the sender observes a terminal node $z_t \in \mathcal{Z}$ that uniquely determines a path in the tree defining the SDM problem. We encode such a path by means of a vector $\boldsymbol{p}_t \in \{0,1\}^{|\Sigma_c|}$

---

[10]See Lemma 9 and 10 in Appendix F for the formal statements of these properties.

such that $\boldsymbol{p}_t[\sigma] = 1$ if and only if the chance sequence $\sigma \in \Sigma_c$ lies on the path from the root of the tree to $z_t$, namely $\sigma \preceq \sigma_c(z_t)$. If the sender commits to a signaling scheme $\boldsymbol{\phi}_t \in \Phi$, then it is easy to see that, for every $\sigma = (h, a) \in \Sigma_c$, the element $\boldsymbol{p}_t[\sigma]$ is distributed as a Bernoulli of parameter $\boldsymbol{\phi}_t[\sigma_s(h)]\boldsymbol{\mu}^\star[\sigma]$. The crucial observation behind the design of our estimator is that, if the sender commits to a deterministic signaling schemes $\boldsymbol{\pi}_t \in \Pi$ at some round $t \in [T]$, then for all the chance sequences $\sigma \in \Sigma_c$ that are *compatible* with $\boldsymbol{\pi}_t$, *i.e.*, that can be observed when $\boldsymbol{\pi}_t$ is played, we have that $\boldsymbol{p}_t[\sigma]$ is distributed as a Bernoulli of parameter $\boldsymbol{\mu}^\star[\sigma]$. Formally, a sequence $\sigma \in \Sigma_c$ is compatible with $\boldsymbol{\pi}_t$ if there exists a chance node $h \in \mathcal{H}_c$ and an outcome $a \in A(h)$ satisfying $\sigma = (h, a)$ and $\boldsymbol{\pi}_t[\sigma_s(h)] = 1$. This observation leads to the following result:

**Lemma 3.** *For every deterministic signaling scheme $\boldsymbol{\pi} \in \Pi$, let*

$$\Sigma_\downarrow(\boldsymbol{\pi}) := \{\sigma = (h, a) \in \Sigma_c \mid a \in A(h) \wedge \boldsymbol{\pi}[\sigma_s(h)] = 1\}.$$

*Then, during each round $t \leq N$ of Algorithm 2, it holds $\mathbb{E}[\boldsymbol{p}_t[\sigma]] = \boldsymbol{\mu}^\star[\sigma]$ for every $\sigma \in \Sigma_\downarrow(\boldsymbol{\pi}_t)$.*

Thus, during the first phase, Algorithm 2 builds the desired estimator $\widehat{\boldsymbol{\mu}}_N$ of $\boldsymbol{\mu}^\star$ as follows. At each round $t \leq N$, after observing the feedback $z_t$, the algorithm samples a deterministic signaling scheme $\boldsymbol{\pi}_t \in \Pi$ according to $\boldsymbol{\phi}_t$ (the one actually selected at $t$), so that all the sequences $\sigma \in \Sigma_c$ such that $\boldsymbol{p}_t[\sigma] = 1$ (or, equivalently, $\sigma \preceq \sigma_c(z_t)$) belong to $\Sigma_\downarrow(\boldsymbol{\pi}_t)$.[11] Then, for every $\sigma \in \Sigma_\downarrow(\boldsymbol{\pi}_t)$, the algorithm updates the estimator component $\widehat{\boldsymbol{\mu}}_t[\sigma]$ according to $\boldsymbol{p}_t[\sigma]$. Since the probability of visiting a sequence $\sigma \in \Sigma_c$ depends on $\boldsymbol{\phi}_t$ (and, thus, can be arbitrarily small), the first $N$ rounds must be carefully used to ensure that each sequence is explored at least $N/|\Sigma_c|$ times. To explore a specific sequence $\sigma \in \Sigma_c$, we choose a signaling scheme $\boldsymbol{\phi}_t$ such that $\sigma \in \Sigma_\downarrow(\boldsymbol{\pi}_t)$ for every deterministic $\boldsymbol{\pi}_t \sim \boldsymbol{\phi}_t$. The procedure described above is needed for minimizing the receiver's regret, since, in the second phase, the algorithm selects signaling schemes $\boldsymbol{\phi}_t$ from $\Lambda_{\beta_N}(\widehat{\boldsymbol{\mu}}_N)$. In particular, as shown by the following lemma, Algorithm 2 guarantees that the receiver's regret is upper bounded by $2\beta_N$ at each round $t > N$, since it defines $\epsilon_t[\sigma]$ for each sequence $\sigma \in \Sigma_c$ so that the event $\tilde{\mathcal{E}} := \{|\boldsymbol{\mu}^\star[\sigma] - \widehat{\boldsymbol{\mu}}_t[\sigma]| \leq \epsilon_t[\sigma] \ \forall (t, \sigma) \in [T] \times \Sigma_c\}$ holds with probability at least $1 - \delta/2$.

**Lemma 4.** *Under the event $\tilde{\mathcal{E}}$, Algorithm 2 guarantees that $\boldsymbol{\phi}_t \in \Lambda_{2\beta_N}(\boldsymbol{\mu}^\star)$ at each round $t > N$.*

### 7.2 Minimizing the sender's regret

Algorithm 2 also needs to guarantee small sender's regret. To do so, we would like that $\boldsymbol{\phi}^\star$ is a valid pick for the algorithm, *i.e.*, it belongs to $\Lambda_{\beta_N}(\widehat{\boldsymbol{\mu}}_t)$. However, differently from the full-feedback setting, stopping exploration after the first $N$ round does *not* guarantee optimal rates. In order to fix this issue, in the second phase, the algorithm selects $\boldsymbol{\phi}_t$ optimistically by maximizing the sender's expected utility $F(\boldsymbol{\phi}, \boldsymbol{\mu})$ over both $\boldsymbol{\phi} \in \Lambda_{\beta_N}(\widehat{\boldsymbol{\mu}}_N)$ and $\boldsymbol{\mu} \in \mathcal{C}_t(\delta)$, where $\mathcal{C}_t(\delta)$ is a suitably-defined confidence set centered around $\widehat{\boldsymbol{\mu}}_t$ such that $\{\boldsymbol{\mu}^\star \in \mathcal{C}_t(\delta)\} \equiv \tilde{\mathcal{E}}$, and, thus, it holds with high probability. This guarantees that $\max_{\boldsymbol{\mu} \in \mathcal{C}_t(\delta)} F(\boldsymbol{\phi}^\star, \boldsymbol{\mu}) \geq F(\boldsymbol{\phi}^\star, \boldsymbol{\mu}^\star)$. Formally:

**Lemma 5.** *If the event $\tilde{\mathcal{E}}$ holds, then, for every round $t > N$, it holds that $\boldsymbol{\phi}^\star \in \Lambda_{\beta_N}(\widehat{\boldsymbol{\mu}}_t)$ and $\max_{\boldsymbol{\mu} \in \mathcal{C}_t(\delta)} F(\boldsymbol{\phi}^\star, \boldsymbol{\mu}) \geq F(\boldsymbol{\phi}^\star, \boldsymbol{\mu}^\star)$.*

Thus, $F(\boldsymbol{\phi}_t, \boldsymbol{\mu}^\star) \approx F(\boldsymbol{\phi}_t, \widehat{\boldsymbol{\mu}}_t) \geq \max_{\boldsymbol{\mu} \in \mathcal{C}_t(\delta)} F(\boldsymbol{\phi}^\star, \widehat{\boldsymbol{\mu}}) \geq F(\boldsymbol{\phi}^\star, \boldsymbol{\mu}^\star)$ holds in the limit, implying that $F(\boldsymbol{\phi}_t, \boldsymbol{\mu}^\star)$ converges to $F(\boldsymbol{\phi}^\star, \boldsymbol{\mu}^\star)$ after sufficiently many rounds. Formally:

**Theorem 5.** *Given any $\delta \in (0, 1)$ and $N \in [T]$, Algorithm 2 guarantees:*

$$R_T = \mathcal{O}\left(N + \sqrt{\log\left(\frac{T|\Sigma_c|}{\delta}\right)|\Sigma_c|T}\right) \quad and \quad V_T = \mathcal{O}\left(N + T|\mathcal{Z}|\sqrt{\log\left(\frac{T|\Sigma_c|}{\delta}\right)\frac{|\Sigma_c|}{N}}\right),$$

*with probability at least $1 - \delta$. Moreover, the algorithm runs in polynomial time.*

In contrast to the case with full feedback, the optimization problem solved by Algorithm 2 belongs to the class of bilinear problems, which are NP-hard in general [Hillar and Lim, 2013]. However, in Theorem 5 we prove that our specific problem can be solved in polynomial time. Furthermore, notice that Theorem 5 takes as input the number $N$ of rounds devoted to the first phase. Given an $\alpha \geq 1/2$, by choosing any $N = \lfloor T^\alpha \rfloor$ we get bounds of $R_T = \tilde{\mathcal{O}}(T^\alpha)$ and $V_T = \tilde{\mathcal{O}}(T^{\max\{\alpha, 1 - \frac{\alpha}{2}\}})$.

---

[11]The sampling of $\boldsymbol{\pi}_t \in \Pi$ according to $\boldsymbol{\phi}_t$ can be done efficiently by a straightforward modification of the recursive procedure in Farina et al. [2021a,b].

### 7.3 The lower bound frontier

We conclude by showing that the trade offs between $V_T$ and $R_T$ achieved by Algorithm 2 are essentially tight. Previously, we provided an intuition as to why the algorithm needs to uniformly explore the entire tree of the SDM problem. Here, we provide a lower bound that corroborates such a statement. In particular, the following theorem shows that, for any $\alpha \in [1/2,\ 1]$, in order to guarantee a sender's regret of the order of $\mathcal{O}(T^\alpha)$, it is necessary to suffer a receiver's regret of the order of $\Omega(T^{1-\alpha/2})$.[12]

**Theorem 6.** *For any $\alpha \in [1/2, 1]$, there exists a constant $\gamma \in (0,1)$ such that no algorithm guarantees both $R_T = o(T^\alpha)$ and $V_T = o(T^{1-\alpha/2})$ with probability greater than $\gamma$.*

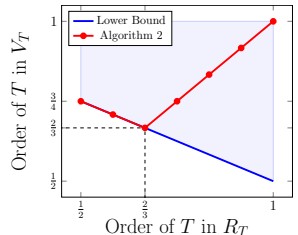

Figure 1: Trade-off between $R_T$ and $V_T$ in the bandit feedback.

Figure 1 shows on the horizontal axis the order of the $T$ term in $R_T$, while, on the vertical axis, it shows the order of the $T$ in $V_T$. The shaded area over the blue line shows the achievable trade offs, while the marked red line shows the performances proved in Theorem 5. Thus, we show that Algorithm 2 matches the lower bound for $\alpha \in [1/2, 2/3]$. However, when $\alpha \in [2/3, 1]$, the guarantees proved in Theorem 5 diverge from the ones proved in the lower bound. This is due to the $N = \lfloor T^\alpha \rfloor$ component in the receiver's regret that becomes dominant when $\alpha \geq 2/3$. We conjecture that it is possible to reduce this term to $\sqrt{N}$, hence matching the lower bound of Theorem 6. The reason for such a gap between the lower and upper bounds is that, during the first phase, Algorithm 2 utilizes signaling schemes without taking into account their persuasiveness, thus incurring in large receiver's regret during the first steps. We leave as future work addressing the question of whether it is possible to design exploration strategies by only using approximately-persuasive signaling schemes.

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
