# OpenReview forum: "Sequential Information Design: Learning to Persuade in the Dark"
_NeurIPS.cc/2022/Conference — NeurIPS 2022 Accept_

### Official Review · Reviewer_tcSK · 2022-07-06

**Rating:** 7
**Confidence:** 3
**Soundness:** 3 good
**Presentation:** 3 good
**Contribution:** 3 good

**Summary:**

The authors study the problem of Bayesian persuasion (BP) with an unknown prior in the sequential decision making (SDM) problem. The traditional BP problem is as follows: a sender with knowledge of the outcomes of random events commits to a signaling scheme (i.e., a mapping from random states to action recommendations) before observing the outcome of the random event. The sender then signals according to their signaling scheme, and the receiver takes an action based on the signal. In the classic BP setting, the goal of the sender is to design an incentive compatible (persuasive) signaling scheme which maximizes their utility in expectation over the random event prior. In addition to being the first to study BP in the SDM setting, the authors consider the case in which the prior over random events is unknown. Since the prior is unknown to the sender, the authors study an online version of the problem in which a sequence of T receivers arrive one after another.

First, the authors provide an (approximate) characterization of the set of persuasive signaling schemes via a polynomially-sized polytope. They show that using this polytope, the sender can compute the optimal signaling scheme by solving a polynomially-sized linear program when the prior is known.

When the prior is unknown, the authors show that it is generally impossible to guarantee that a sequence of signaling schemes are persuasive with high probability. Because of this, they focus on the relaxed goal of ensuring that the cumulative receiver regret is sub-linear in the time horizon.

The authors provide a learning algorithm (Algorithm 1) for the full-feedback setting (i.e., the setting in which the sender is able to observe the realized outcome of all possible random events) which guarantees O(\sqrt{T}) sender and receiver regret with high probability. Next, they build on their algorithm for full feedback to provide an algorithm for the bandit feedback setting (Algorithm 2). Under the bandit feedback setting, the sender only observes the terminal node which is reached at the end of the SDM problem. At a high level, the algorithm is split into two phases: an explore phase, which uses a uniform exploration policy to build an unbiased estimator of the prior, and an exploit phase, which uses the "optimism in the face of uncertainty" priniciple to minimize sender regret. Depending on how long the exploration phase runs for, there is a tradeoff between sender and receiver regret. However, the authors show that this tradeoff is tight for a large portion of the tradeoff frontier.

**Questions:**

What are some concrete examples of a situation in which a sender would have knowledge of the outcomes of the chance nodes of a SDM and would want to persuade a sequence of receivers?

Point of confusion: In the BP literature, there is usually an inherent misalignment between sender and receiver objectives. In the traditional BP setting with a known prior, this issue is addressed by using a signaling scheme which maximizes sender utility subject to the constraint that the receiver is incentivized to follow the sender's action recommendations. In this paper, the authors study a relaxed setting in which the sender wants to minimize receiver regret. Intuitively, why is there no tradeoff between sender and receiver regret, when this incentive mismatch between sender and receiver still exists? It feels like there should be some tension between minimizing sender and receiver regret, similar to the tradeoff which arises due to the exploration phase in Algorithm 2.

**Limitations:**

The authors adequately address the limitations of their work.

**Strengths And Weaknesses:**

Strengths:
The authors study a novel and mathematically interesting version of the Bayesian persuasion problem and provide a relatively complete set of results. Additionally, the presentation of the results is clear, and the technical details appear to be correct.

Weaknesses:
The only real weakness I see is the lack of motivation. In the introduction, the authors motivate the general BP problem, but not the specific SDM setting they consider. Therefore, it is hard to gauge the significance of the authors' results. On one hand, if the problem of BP in SDMs can be well-motivated, I believe this paper is certainly worthy of acceptance. On the other hand, if the setting is not a realistic one in which persuasion may occur, the results may be mathematically interesting, but not very significant. For this reason I have given this submission a borderline accept, although I would be more than happy to raise my score if the authors can address my questions listed below.

---

> ### Author Response · Authors · 2022-08-02
> **Response to Reviewer tcSK**
>
> We thank the Reviewer for the insightful comments about the paper.
>
> **On the applicability of our model.** Bayesian persuasion problems in which the sender and the receiver intercat sequentially have received a lot of attention during recent years, and there are plenty of works on the topic that offer an extensive overview of possible application scenarios. For instance, possible application scenarios range from sequential auctions [Leme et al. 2012] to online navigation apps [Yang et al. 2019]. We agree with the Reviewer that exhibiting a concrete application example would be helpful in furthering motivating the relevance of our sequential Bayesian persuasion model.
> We describe in the following a concrete application example in the context of an online navigation app. The design of a route recommendation policy in a navigation app encompasses all the elements in our model: the sender (the app) receives information about traffic condition sequentially (the traffic changes over time, especially for long routes) and can send recommendations to the receiver (the driver) at at any point in time; the receiver receives information sequentially and takes action sequentially (at any point in time time, it can choose a different route); the receiver is non-myopic. Moreover, the receiver interacts with the navigation app multiple times and, thus, the application must have a small receiver’s regret, otherwise the receiver will decide to move to another navigation app. Let us remark that, in such an application context, it is of paramount importance to consider non-myopic receivers, as one is interested in minimizing the cost of the entire route rather than taking greedy directions at every node.
>
> Leme, Renato Paes, Vasilis Syrgkanis, and Éva Tardos. "Sequential auctions and externalities." Proceedings of the twenty-third annual ACM-SIAM symposium on Discrete Algorithms. Society for Industrial and Applied Mathematics, 2012.
>
> Pu Yang, Krishnamurthy Iyer, Peter I. Frazier. “Information Design in Spatial Resource Competition.” WINE 2019: 346
>
>
> **Tradeoff between sender and receiver regret.** Sublinearity of the receiver’s regret should not be interpreted as an interest of the sender in maximizing the receiver’s utility but as a relaxation of the classical persuasiveness constraints. Indeed, having a small receiver’s regret is equivalent to relaxing the persuasiveness constraints by requiring that they are satisfied asymptotically. Furthermore, we remark that the tradeoff between the receiver’s utility and the sender’s utility is still present in our model, as the Reviewer correctly pointed out. However, this “tension” is inherently present in the definition of an optimal signaling scheme, as in the classic Bayesian persuasion problem. The receiver’s and sender’s regret bounds of our algorithms are insteed related to the problem of \emph{learning} the unknown prior. We show that in the full-information setting there is no tradeoff in \emph{learning} an optimal signaling scheme as we can guarantee the same regret bound for both the sender and the receiver. Indeed, we show that we obtain an optimal bound even if one wants to minimize the sender’s or the receiver’s regret alone (see answer to Review **aKi3** for a lower bound). On the other hand, such tradeoff is present in the bandit-feedback setting, where exploration is not free. Indeed, one can think about the exploration phase of Algorithm 2 as the part which takes care of the receiver's regret, while the second phase takes care of the sender's regret. We think that this is an important discussion and we will add it in the final version of the paper.

---

> > ### Comment · Reviewer_tcSK · 2022-08-03
> > **Reply to authors**
> >
> > Thanks for your reply. Your response about the tradeoff between sender and receiver regret makes sense, as well does your concrete example. Since my main points of contention have been addressed, I will raise my score from 5 to 7. If the paper is accepted, I would encourage the authors to make use of this example throughout the text.

---

### Official Review · Reviewer_xoxk · 2022-07-11

**Rating:** 7
**Confidence:** 3
**Soundness:** 3 good
**Presentation:** 3 good
**Contribution:** 4 excellent

**Summary:**

The paper studies a Bayesian persuasion problem in a setting in which the sender and receiver do not initially know the distribution/prior over random events, and must learn this distribution over time. The goal is to provide algorithms that are low regret for the receiver in this setting (assuming that the sender follows the recommendation) while guaranteeing that it is low regret for the sender to follow the recommendations of the algorithm.

**Questions:**

- Could there be other strategies for the receiver that yield similar or better regret, and is the author’s algorithm robust to such a chance in receiver behavior?



**Limitations:**

The paper does discuss the limitations of their current regret bounds and how they only seem tight for \alpha < 2/3. The paper does not directly discuss potential harms, but I’m not sure whether the discussion of such harms is particularly relevant in a theory paper about Bayesian persuasion (it might be more suited to a paper that applies these techniques to a problem of practical interest).

**Strengths And Weaknesses:**

Strengths:
- The authors consider a novel setting for signaling where the sender and receiver do not know the distribution of state of the world/random events in advance. I think this is an interesting and refreshing contrast from the standard (and often unrealistic) assumption that there is a commonly known prior
- It is nice that the authors consider two types of feedback: a more informative full-feedback setting in which all random events are observed, versus a bandit-style feedback where we only receive information about the arms that we actually pull.
- The results feel tight and complete: the paper both develops a low regret algorithm for persuasion but also show that the trade-off achieved by their algorithm is essentially tight/cannot be improved for non-trivial regimes of parameters \alpha.

Weaknesses:
- The writing feels sometimes too formal and technical. I understand the authors want to give proper preliminaries for sequential decision making problems, but the formalism involved in describing SDMs in all their generality is heavy, and how it maps to/what the chance and decision nodes represent in the specific problem at hand only comes up on page 4. I think in this case it is avoidable and the main body of the paper can be written almost entirely in the simpler language of Section 3. Please let me know if I am wrong, but here it seems the SDM language may really only needed when it comes to showing technical results and could be moved to the appendix.
- Back to the formalism, there are places in the paper where it could really benefit from some English/intuitive descriptions of some of the definitions. E.g., there are two very similar definitions of deviation policies with little explanation of how they differ from each other (I do understand after looking carefully at the definitions that the first one allows for deviations at several points/infosets of the algorithm while the single-point deviation is less general and can only deviate at a single, pre-specified infoset, but given the heavy-handed notations and formalism, this is a bit hard to parse)
- The paper shows that it is sublinear regret for the receiver to follow the algorithms recommendations, showing that it is a good strategy for the receiver. It however is not clear to me that it implies that this is the only good strategy for the receiver, and the regret guarantees for the sender assumer the receiver always follows the recommendation. I think it might be worth highlighting and discussing this point a bit more, and it would be great (though probably out of scope) to have an algorithm for the sender that is robust to any no-regret algorithm the receiver may be using.

Overall I think this is i) an interesting problem to look at where the sender and receiver do not have prior information about the distribution of “states of the world” and ii) comes with good regret bounds for both the sender and receiver with tight bounds for the sender, and so I believe the technical contribution of the paper warrants acceptance. However, I think clarity can be improved significantly here and the writing is really geared towards experts in sequential decision making and extensive form games and possibly a bit hard to understand otherwise.

---

> ### Author Response · Authors · 2022-08-02
> **Response to Reviewer xoxk**
>
> We thank the Reviewer for the positive comments about the paper and the interesting feedback provided.
>
> **On the SDM language.** Introducing the SDM language in the main paper is necessary in order to present one of our main contributions, which is the characterization of persuasive signaling schemes and the resulting polytopal approximation, described in Section 4. We believe that the existence of a linear programming formulation of our sequential Bayesian persuasion problem is one of the main contributions of the paper. Indeed, this is the first result showing how to compute in polynomial time a sender’s optimal policy in sequential Bayesian persuasion problems with a farsighted receiver. This is in contrast with what happens in MDPs (see [17] and the answer to reviewer **aKi3**). On the other hand, we agree with the Reviewer that some definitions may require more intuition and we will expand some of the explanations of the technical definitions in the final version of the paper.
>
> **On the existence of multiple receiver’s optimal strategies.** Similarly to most of the Bayesian persuasion literature, we need to assume that the receiver follows the sender’s recommendation when she is (almost) indifferent between multiple strategies. However, we show that the recommended strategy has no regret with respect to the optimal deviation policy, and the optimal deviation policy is the optimal receiver’s strategy. Thus, it clearly establishes an upper bound of the utility achievable by any regret minimizer. Hence, we can guarantee that following the sender’s recommandations has no regret with respect to using any regret minimizer.

---

> > ### Comment · Reviewer_xoxk · 2022-08-07
> > **Thanks a lot for the response & sorry for the delay**
> >
> > Thanks a lot for the response! Regarding i) and re-looking at it carefully, I agree that it would seem hard to explain the main theorem without the SDM language. It'd be great if you had a bit of space to give some intuitive explanation of some of the definitions to make it a bit easier to read, there are quite a lot of notations to parse through.
> >
> > I agree with point ii; thanks for the clarification!
> >
> > I'll up my score to a 7; I was a bit worried about some of the presentation, but I still think it is a good paper, I am overall satisfied with the authors' answers, and I think adding a bit of an explanation in the camera-ready should make the paper easy to read. Thanks again!

---

### Official Review · Reviewer_xRVr · 2022-07-12

**Rating:** 6
**Confidence:** 4
**Soundness:** 3 good
**Presentation:** 2 fair
**Contribution:** 3 good

**Summary:**

The paper studies Bayesian persuasion between an informed sender and a
receiver in a sequetial decision making (SDM) setting. In contrast to
standard assumptions in the literature, the sender is uninformed of
the prior of the payoff relevant state, whereas the receiver is
farsighted. The paper designs an online-learning algorithm that
attains no-regret for the sender (against the known-prior benchmark)
while being persuasive for the receiver, where the latter criterion is
defined as a low-regret condition. The paper considers both the
full-feedback setting (where the sender observes all the random
realizations in the SDM decision tree, even off-path), and the bandit
feedback setting (where only the random realizations on the traversed
path is observed by the sender). For the full-feedback setting, the
authors provide an algorithm that achieves $\tilde{O}(\sqrt{T})$
regret for both the sender and the receiver. For the bandit-feedback
setting, the regret guarantees are somewhat weaker for both the sender
and the receiver.

**Questions:**

It might be helpful if the authors can motivate the repeated SDM
persuasion problem through an application context. I don't expect an
exact fit, but some examples would help in highlighting the
significance of the problem.

Is there a lower-bound characterization in the full-feedback setting?

Repeating the comment above, in the full-feedback setting, if the
sender can observe the *past* realizations on off-path chance nodes,
*and* the randomness across chance nodes that are not
descendant-ancestor pairs are allowed to be correlated, then observing
such *past* realizations can be beneficial to the sender. Does this
affect the analysis?

**Strengths And Weaknesses:**

The paper considers an extension of Zu et al (2021) who consider the
problem of learning to persuade a receiver taking a single action at
each time, when prior distribution is unknown. Here, the authors study
this problem in settings where the receiver has to make multiple
sequential decisions at each time period (and the sender makes
multiple recommendations, at each information set of the receiver).
Unlike Zu et al, the recommendations here need not be persuasive (and
the authors show that it may not be possible to achieve this); instead
the authors require that the receiver incurs low regret from following
the sender's recommendations. The authors propose a natural signaling
algorithm that achieves this notion of persuasiveness while at the
same time ensuring that the sender's regret is low.

While the theoretical contribution is solid, the paper does not
motivate the specific applications of their model. In particular, the
model requires that a single sender and a single receiver engage in
repeated sequential decision making (with iid noise) -- it is unclear
to me where such an interaction might arise in an application. One may
motivate the repeated interaction as between a single sender and a
stream of receivers (as in Zu et al), but then the no-regret version
of persuasiveness for the receiver needs justification.

Furthemore, the modeling assumptions about the sender's knowledge are
unclear. For instance, I believe the paper assumes that the sender,
when making a recommendation, observes the specific decision node of
an information set reached by the receiver, but not the realizations
of future randomness. It would be helpful to make this explicit. But
more importantly, in the full-feedback setting, if the sender can
observe the *past* realizations on off-path chance nodes, *and* the
randomness across chance nodes that are not descendant-ancestor pairs
are allowed to be correlated, then observing such *past* realizations
can be beneficial to the sender.

Relatedly, one would also need to make further assumptions about the
sequence form prior, as current formulation may allow correlations at
chance nodes that are not decendant-ancestor of each other. As
mentioned above, this may be important for the full-feedback setting.

---

> ### Author Response · Authors · 2022-08-02
> **Response to Reviewer xRVr**
>
> We thank the Reviewer for the positive comments about the paper and the interesting feedback provided.
>
> **On the applications of our model.**  Bayesian persuasion problems in which the sender and the receiver intercat sequentially have recently received a lot of attention, since they enable the application of Bayesian persuasion to a wider spectrum of new settings, ranging from sequential auctions and online shopping platforms, to online navigation apps. We refer the Reviewer to the answer to Reviewer **tcSK** for a more detailed discussion on this.
>
> **On the knowledge of the sender.** We agree with the reviewer that we could remark better that the sender only observes the feedback at the end of every turn $t$ (see the beginning of Section 3) and thus does not observe future realization of random events. The sender can only observe the current node during the sequential game and sample from the signaling scheme $\phi_t$ to which the sender has already committed to. We will insert this remark in the final version of the paper.
>
> **On the feedback in the full information setting.** In our work, we assumed that there is no correlation among the events at chance nodes. This means that an outcome is sampled at every chance node independently from the outcomes sampled at other chance nodes, which allows us to represent the prior by using the sequence form. However, we agree with the Reviewer that considering correlation between events at chance nodes that are not descendant and ancestor of each other could be an interesting extension to our model. In such a setting, we agree with the Reviewer that the sender can obtain further information from the observed history of outcomes sampled at chance events.
>
> **On the lower bound for the full-feedback setting.** See the answer to Reviewer **aKi3**.

---

> > ### Comment · Reviewer_xRVr · 2022-08-07
> > **response**
> >
> > Thanks for the detailed response, and the clarifications on the questions.

---

### Official Review · Reviewer_aKi3 · 2022-07-13

**Rating:** 7
**Confidence:** 2
**Soundness:** 3 good
**Presentation:** 3 good
**Contribution:** 3 good

**Summary:**

This paper studies bayesian persuasion where a farsighted receiver plays in a sequential decision making problem, and considers both full feedback and bandit feedback setting. The authors first proved a negative result that shows it is impossible to be persuasive without the knowledge of the prior. Then a relaxed online learning problem is considered, and the authors proposed algorithms that guarantee sub-linear regret for both the sender and the receiver.


**Questions:**

I'd appreciate it if the authors could help me interpret the result in this paper compared with the negative result for farsighted receiver in MDP in [17]. What is the key difference between the two problem settings that makes the problem in [17] NP hard, while the problem in this paper solvable in polynomial time?

Is the trade-off between the regret of sender and receiver attained by Algorithm 1 optimal for the full feedback setting?

**Limitations:**

No.

**Strengths And Weaknesses:**

The problem of bayesian persuasion in sequential problem is well-motivated. However, prior works [17, 25] only consider myopic user. This paper is the first to propose algorithms for farsighted receiver in sequential decision making problem, which can be solved in polynomial time.

---

> ### Author Response · Authors · 2022-08-02
> **Response to Reviewer aKi3**
>
> We thank the Reviewer for the positive comments about the paper and the interesting feedback provided.
>
> **On the relation with the results in [17].** Our model and the one presented in [17] are somehow related, even though they have some crucial differences. Indeed, any finite-horizon MDP can be cast into our SDM model. However, from a computational perspective the two models are incomparable: our model is richer (since it offers the possibility of modeling partial observability of the MDP states), but it offers a less compact representation. In particular, an MDP can be reduced to our model with a (potentially) exponential blowup of the representation size. The crucial difference between our model and the one in [17] is that our model allows us to consider non-Markovian policies, while [17] only considers Markovian policies. Thus, a possible interpretation of our results is the following: we show that non-Markovian policies are a possible intriguing possibility when it comes to computing optimal persuasive policies in MDPs. We leave pursuing this interesting research direction to future works. We will add additional comments on the relation between our model and that in [17] in the final version of the paper.
>
> **On the regret bound with full feedback.** The regret bound attained by Algorithm 1 is optimal. Indeed, a reduction to the expert problem provides a $ \Omega(\sqrt{T})$ lower bound on both the sender’s regret and the receiver’s regret. In particular, the reduction works by setting the receiver's and the sender’s utilities to be equal and by considering an instance in which the agent takes only one decision and has $n$ actions available. After each one of the $n-1$ actions, chance acts by selecting either an outcome with a utility of $1$ or one with a utility of $0$, with a resulting expected value of $1/2$, while the $n$-th action results in an expected value of $1/2+\epsilon$ for the receiver. This is indeed the same instance as the one used to prove a lower bound of $ \Omega(\sqrt{T})$ for the regret in classical expert problems. In order to prove our lower bound for the bandit feedback setting in Theorem 6, we used analogous techniques. We will add a detailed remark on this fact in the final version of the paper.

---

### Meta-Review · Area_Chair_RXx9 · 2022-08-21

**Recommendation:** Accept
**Confidence:** Certain

**Metareview:**

The reviews are all positive and reviewers agree that the paper studies a novel setting with solid technical contributions.

**Award:**

No

---

### Decision · Program_Chairs · 2022-09-14

Accept